# *Clonorchis sinensis* infection modulates key cytokines for essential immune response impacted by sex

Shuo Kan[1,2☯], Qi Li[1,2☯], Hong-Mei Li[3], Yan-Hua Yao[4], Xin-Yue Du[2], Chen-Yun Wu[2], Guang-Jie Chen[2], Xiao-Kui Guo[1], Men-Bao Qian[1,3]*, Zhao-Jun Wang[1,2]*

**1** NHC Key Laboratory of Parasite and Vector Biology; School of Global Health, Chinese Center for Tropical Diseases Research-Shanghai Jiao Tong University School of Medicine, Shanghai, China, **2** Shanghai Institute of Immunology, Department of Immunology and Microbiology, Shanghai Jiao Tong University School of Medicine, Shanghai, China, **3** National Institute of Parasitic Diseases, Chinese Center for Disease Control and Prevention (Chinese Center for Tropical Diseases Research); WHO Collaborating Centre for Tropical Diseases; National Center for International Research on Tropical Diseases, Shanghai, China, **4** Department of Biochemistry and Molecular Cell Biology, Shanghai Jiao Tong University School of Medicine, Shanghai, China

☯ These authors contributed equally to this work.
* qianmb@nipd.chinacdc.cn (MBQ); zjwang@sjtu.edu.cn (ZJW)

**Editor:** jong-Yil Chai, Seoul National University College of Medicine, REPUBLIC OF KOREA

**Data Availability Statement:** All relevant data are within the manuscript and its Supporting information files.

## Abstract

Infection with helminths can modulate the host immune response, which ultimately shape morbidity and mortality of the associated diseases. We studied key cytokines for essential immune response in sera from 229 southeastern China individuals infected with *Clonorchis sinensis* and 60 individuals without *C. sinensis* infection, and measured serum specific IgG and IgE against worms in these people. Individuals infected with *C. sinensis* had significantly higher antigen-specific IgG and IgE levels, which were positively correlated with egg counts in feces. However, less enhancement of IgE antibody was observed in females when compared to males with similar infection levels. *C. sinensis* infection caused diminished Th1 cytokines (IL-1β, IL-2, IL-12p70, IFN-γ and TNF-α), Th2 cytokine (IL-4), as well as Th17 cytokine (IL-17A) in sera, which showed decreasing trend by infection intensity. Notably, these phenotypes were more significant in females than those in males. Although *C. sinensis* infection is associated with the development of hepatobiliary diseases, there was no significant correlation between the dampened cytokine profiles and the hepatobiliary morbidities. Our study indicates *C. sinensis* infection is strongly related to the immune suppression in human. Sex differences shape the immune milieus of clonorchiasis. This study provides a better understanding of how worms affect immune responses and cause a long-term immune alternation in humans with *C. sinensis* infection.

## Author summary

*Clonorchis sinensis*, also known as the liver fluke, lives in human bile duct system and is endemic in East Asia. Chronic *C. sinensis* infection without treatment can result in serious

**Funding:** ZJW was supported by the National Natural Science Foundation of China (81971486), the Natural Science Foundation of Shanghai (19ZR1428500), and the Fifth Round of Three-Year Public Health Action Plan of Shanghai (No. GWV-10.1-XK13). The funders had no role in study design, data collection and analysis, decision to publish, or preparation of the manuscript.

**Competing interests:** The authors have declared that no competing interests exist.

illness and predispose the human to bile duct cancer. Helminth infection is able to modulate the host immune response and influence the outcome of infection, but the immune characteristics of *C. sinensis* infection is not yet known. In this study, we analyzed serum samples from individuals living in endemic areas with clonorchiasis in China. We found *C. sinensis* infection caused increased specific IgG and IgE to adult worm antigens, but diminished levels of key cytokines for essential immune response. Th1 cytokines (IL-1β, IL-2, IL-12p70, IFN-γ and TNF-α), Th2 cytokine (IL-4), as well as Th17 cytokine (IL-17A) showed decreasing trend by infection intensity. Moreover, females exhibited more significant cytokine variation compared to males with similar infection intensity. Our study indicates that *C. sinensis* infection is related to immune suppression in human, which might contribute to the outcome of clonorchiasis. The sexual dimorphism needs to be considered in the clonorchiasis prophylaxis and immune investigation.

## Introduction

Parasitic worms coexist with human beings for a very long time. In a life history, host and parasite continually adapt to each other, thus a finely tuned balance between host immunity and chronic parasitism has been developed [1,2]. It has been believed that type-2-cell-mediated immune responses play a critical role in immune responses to parasitic worms [3]. Meanwhile, many parasite species can induce IL-10 production and Treg cell development, then redirect, suppress, and evade host immunity to establish chronic infection [1,4]. Notable progress has been made in understanding helminth immunology, which contributes to disease prophylaxis and immune system investigation. However, most of those advances focus on intestinal helminth or using murine models for intestinal nematodes, such as *Trichuris muris*, *Nippostrongylus brasiliensis*, and *Heligmosomoides polygyrus* [5]. Various parasites with different stages reside in different tissue locations during their life cycle. For example, most trematodes are tissue-dwelling helminths [6]. Their roles in immune regulation may not the same as intestinal helminths.

   Clonorchiasis is a food-borne parasitic disease, caused by *Clonorchis sinensis*, which is predominantly endemic in East Asia, including China, Korea and Vietnam [7–9]. About 15 million people are estimated to be affected by this disease. Particularly, about 13 million cases distribute in China [10,11]. The adult worms of *C. sinensis* living in the biliary tree of the liver produce eggs which are passed in feces. *C. sinensis* infection predominantly leads to hepatobiliary abnormalities, such as periductal fibrosis, cholangitis, cholecystitis and cholelithiasis [12–15]. Moreover, it is classified as "carcinogenic to humans" (Group 1) by the International Agency for Research on Cancer, because of the carcinogenesis in fatal cholangiocarcinoma [16]. Up to now, research on the immunology of human clonorchiasis is inadequate, which hinders our understanding on the pathogenesis. In this study, we screened 289 serum samples from individuals living in southeastern China endemic with clonorchiasis, to reveal a physiology long-term immune response profiles of *C. sinensis* infection in human. Our study indicates *C. sinensis* infection is strongly related to the immune suppression, which is influenced by sex.

## Methods

### Ethics statement

The conducts and procedures were approved by the Ethics Committee of the National Institute of Parasitic Diseases, Chinese Center for Disease Control and Prevention in Shanghai, China

(reference no.2011–005). All individuals or their guardians for those aged <18 years have provided written informed consent. Praziquantel (25 mg/kg, t.i.d, 2 days) and albendazole (400mg, single oral dose) were provided to individuals infected with *C. sinensis* for free.

## Human sera

Serum samples were collected from villagers living in Hengxian County, Guangxi, China, where clonorchiasis is highly endemic. Detection of helminth infection, collection of sera, and ultrasound examination for hepatobiliary abnormalities were implemented, as described previously [12,17]. In brief, one stool sample was collected and examined by the Kato-Katz method and washing sedimentation technique. The eggs per gram of feces (EPG) is calculated by multiplying the average of three Kato-Katz smears with a factor of 24. Sera were then collected from the participants and abdominal ultrasound examination was also implemented. A total of 289 participants were included in this study, of which 229 individuals was detected with *C. sinensis* infection (*Cs*+) and another 60 individuals without *C. sinensis* infection (*Cs*-). The characteristics of the participants were summarized in S1 Table.

## Preparation of adult worm antigen of *C. sinensis* (CsAWA)

Adult worm antigen of *C. sinensis* (CsAWA) were prepared for ELISA. In brief, *C. sinensis* adult worms suspended in PBS were homogenized on ice. The mixture was lysed by sonication in an ice-chilled water bath. Then, the lysed homogenate was centrifuged at 15,000 g for 20 min at 4°C. The supernatant was dialyzed against PBS at 4°C overnight and used as CsAWA. Protein concentration was measured by BCA Protein Assay Kit (Sangon Biotech, China).

## Enzyme-Linked Immunosorbent Assay (ELISA)

Antibody reactivity of human sera against CsAWA was determined by enzyme-linked immunosorbent assay (ELISA). Briefly, 96-well plates were coated with 100 μl 2.5 μg/ml CsAWA overnight. Human sera were diluted with 1:100 and HRP conjugated goat anti-human IgG (Sigma-Aldrich, USA, 1:5000 dilution) or IgE (Invitrogen, USA, 1:2000 dilution) was used as the secondary antibody. Next, reactions were developed using 3,3',5,5'-Tetramethylbenzidine (TMB) substrates and stopped with 2 N $H_2SO_4$. The optical densities were read at 450 nm in a microwell reader system (Biotek, USA).

## Serum essential immune response cytokine screen

Key cytokines for essential immune response, including IL-1β, IL-2, IL-4, IL-6, IL-10, IL-12p70, IL-17A, IFN-γ, TNF-α and free active TGF-β1 were measured using a LEGENDplex™ HU Essential Immune Response Panel Kit (BioLegend, San Diego, CA, USA). The data were acquired on a BD FACSCanto™ II flow cytometer, analyzed using LEGENDplex™ Data Analysis Software, and calculated by standard curves according to the manufacturer's instructions.

## Statistical analyses

Statistical analyses were performed using Graphpad Prism 9 and SPSS 24.0 software. The data were presented as the mean ± s.e.m. Statistical significance was analyzed by means of Mann-Whitney U test or Kruskal-Wallis test followed by Dunn's multiple comparisons test. Spearman's correlation analysis was used to analyze the association between the EPG and antibodies or cytokines. The correlation coefficient, r, ranges from -1 to +1. The significance corresponding to the r is: perfect positive correlation r = 1, the two variables tend to increase or decrease together 0 < r < 1, the two variables do not vary together at all r = 0, one variable increases as

the other decreases -1 < r < 0, perfect negative or inverse correlation r = -1. Principal Component Analysis (PCA) was used to extract the main feature components of the data. Results were considered statistically significant difference at $P < 0.05$. The significance corresponding to the asterisk is: $^*P < 0.05$, $^{**}P < 0.01$, $^{***}P < 0.001$.

## Results

### Enhanced levels of IgG and IgE specific antibodies in individuals with *C. sinensis* infection

To determine the immune status of people infected with *C. sinensis*, we firstly measured serum specific antibodies by ELISA. The levels of IgG and IgE antibodies against *C. sinensis* were significantly higher in *Cs+* than in *Cs-* individuals (Fig 1A). Elevated levels of serum IgG and IgE were positively correlated with the intensity of infection expressed as EPG (Fig 1B).

### Less enhancement of specific IgE in female individuals with *C. sinensis* infection

It is well known that sex and age differences shape the immune response to infectious diseases [18]. We analyzed the levels of IgG and IgE in females and males respectively. As shown in Fig 2A, there was a significant difference in serum specific IgG and IgE levels between female and male groups in *Cs+* individuals. Positive correlations were observed between EPG and serum specific IgG and IgE in both female and male groups (Fig 2B). However, less enhancement of IgE antibody was observed in females when compared to males with similar EPG (Fig 2B). We also analyzed the levels of specific antibodies in different age groups (10–29 years, 30–44 years, 45–59 years, and 60–86 years). Elevated levels of serum IgG and IgE were positively correlated with the intensity of infection (EPG) in all four age groups (S1 Fig).

### Diminished serum levels of key cytokines for essential immune response in *C. sinensis* infected individuals

To further determine the effect of *C. sinensis* infection on host immunity, we measured key cytokines for essential immune response in sera, including IL-1β, IL-2, IL-4, IL-6, IL-10, IL-12p70, IL-17A, IFN-γ, TNF-α and TGF-β1. As shown in Fig 3A, *Cs+* individuals had significantly lower levels of IL-1β, IL-2, IL-4, IL-12p70, IL-17A, IFN-γ and TNF-α in comparison with *Cs-* individuals. In addition, the levels of IL-6, IL-10 and TGF-β1 also showed a decreasing trend in *Cs+* individuals compared to *Cs-* individuals. Spearman's correlation analysis indicated that the levels of IL-1β, IL-2, IL-4, IL-10, IL-12p70, IL-17A and TNF-α were significantly decreased with increasing EPG (Fig 3B). Similar trends were found in different age groups (10–29 years, 30–44 years, 45–59 years and 60–86 years groups) (S2 Fig). Overall, *C. sinensis* infection appeared to suppress key cytokines which are associated with innate and adaptive immune responses.

### Sex differences shape the cytokine milieus in *C. sinensis* infection

To study the influence of sex on changes in cytokines, we analyzed the cytokine profiles in females and males respectively. No significant difference was observed between females and males without *C. sinensis* infection. The baseline levels of cytokines were comparable in females and males (S3 Fig). As shown in Fig 4, both females and males had suppressed cytokine milieus post *C. sinensis* infection. There were significant negative correlations between EPG and the levels of IL-1β, IL-4, IL-12p70, IL-17A, TNF-α and TGF-β1 in females. While in male individuals, only IL-17A level showed a significant correlation with EPG. With the

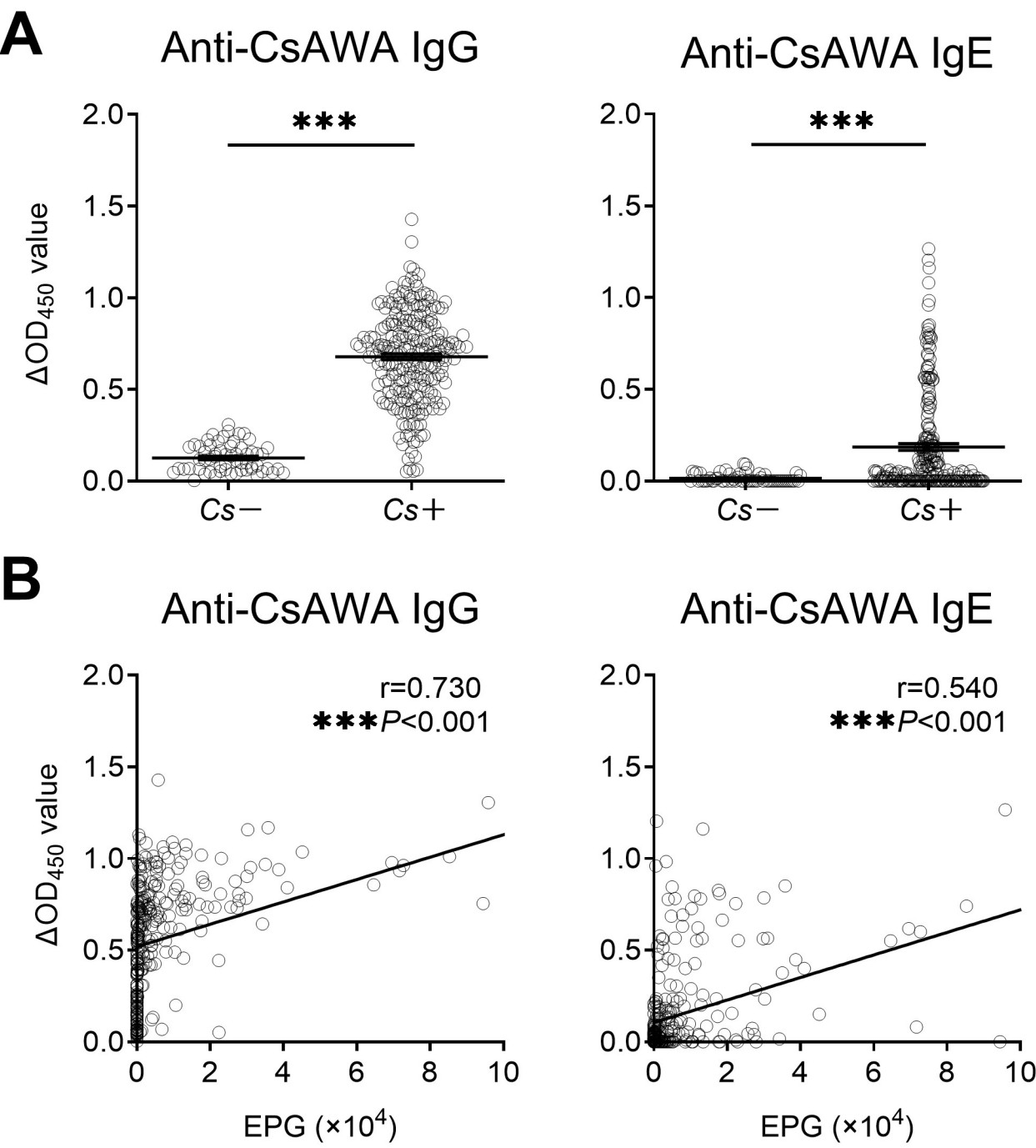

**Fig 1. Serum levels of anti-CsAWA IgG and IgE in *Cs*- and *Cs*+ individuals. (A)** Serum levels of anti-CsAWA IgG and IgE in *Cs*- individuals (n = 60) and *Cs*+ individuals (n = 229). **(B)** Spearman's correlation analysis between EPG and serum levels of anti-CsAWA IgG and IgE (n = 289). CsAWA: *C. sinensis* adult worm antigen, *Cs*: *C. sinensis* infection, EPG: eggs per gram of feces. The data were shown as the mean ± s.e.m., ***P< 0.001, Mann-Whitney U test in **(A)**, Spearman's correlation test in **(B)**.

similar intensity of infection, female individuals showed more reduction in cytokines than male individuals. We further compared the cytokine levels between females and males in different age groups. In general, both females and males had suppressed cytokine milieus post *C. sinensis* infection in 30–44 years, 45–59 years and 60–86 years groups, and female individuals showed more reduction in cytokines than male individuals (S4 Fig).

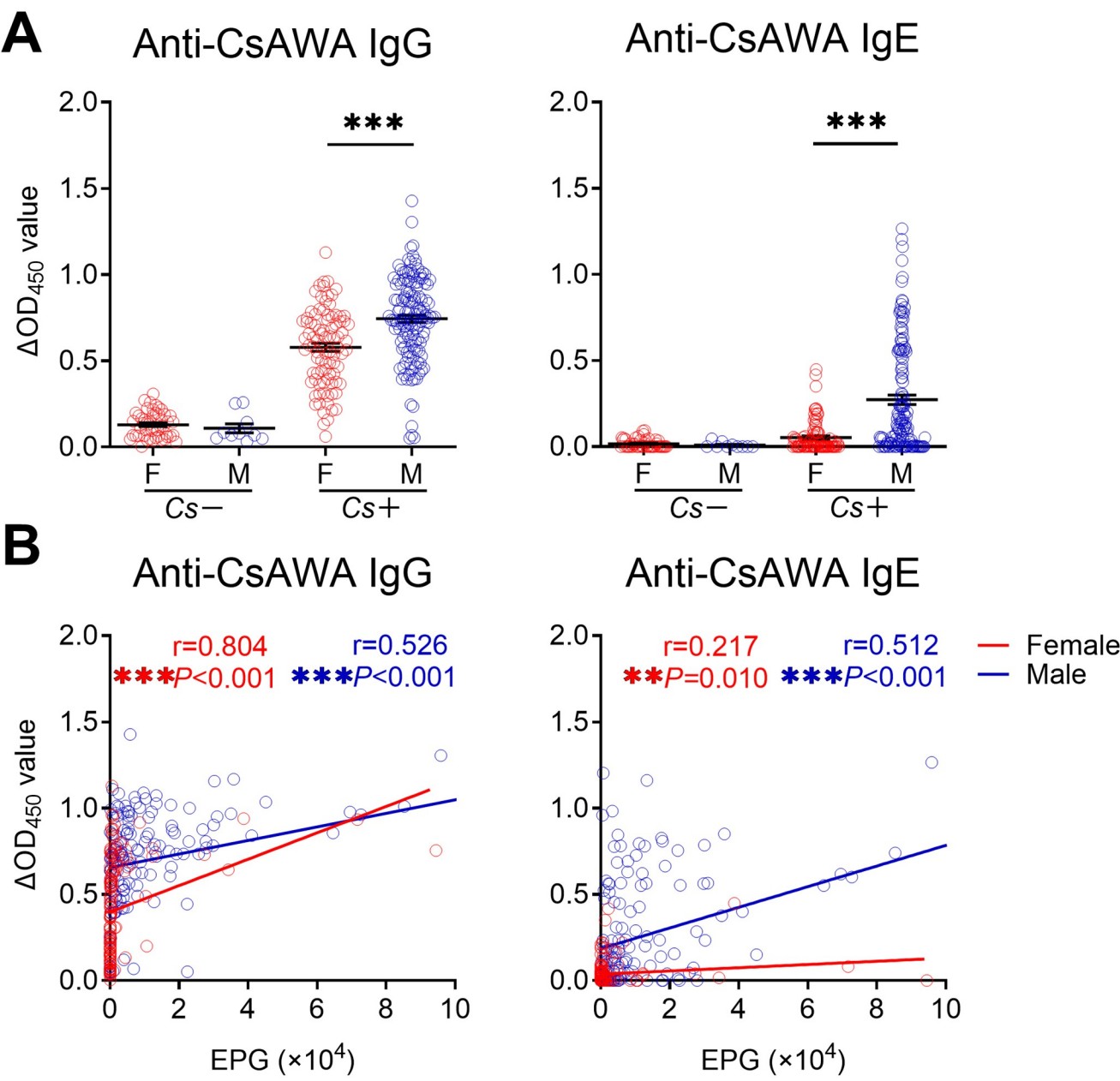

**Fig 2. Serum levels of anti-CsAWA IgG and IgE in *Cs*- and *Cs*+ individuals with different genders. (A)** Serum levels of anti-CsAWA IgG and IgE in *Cs*-individuals (n = 50 in female and n = 10 in male) and *Cs*+ individuals (n = 91 in female and n = 138 in male). **(B)** Spearman's correlation analysis between EPG and serum levels of anti-CsAWA IgG and IgE (n = 141 in female and n = 148 in male). CsAWA: *C. sinensis* adult worm antigen, *Cs*: *C. sinensis* infection, F: female, M: male, EPG: eggs per gram of feces. The data were shown as the mean ± s.e.m., **$P < 0.01$, ***$P < 0.001$, Kruskal-Wallis test followed by Dunn's multiple comparisons test in **(A)**, Spearman's correlation test in **(B)**.

## PCA analysis reveals trends of sex bias in the immune milieu in *C. sinensis* infection

To assess the overall trends in cytokine and antibody discrimination between male and female individuals, we plotted PCA with different inputs. As shown in Fig 5, PCA analysis showed different cytokines and antibodies clusters between *Cs*+ male and female individuals. In contrast,

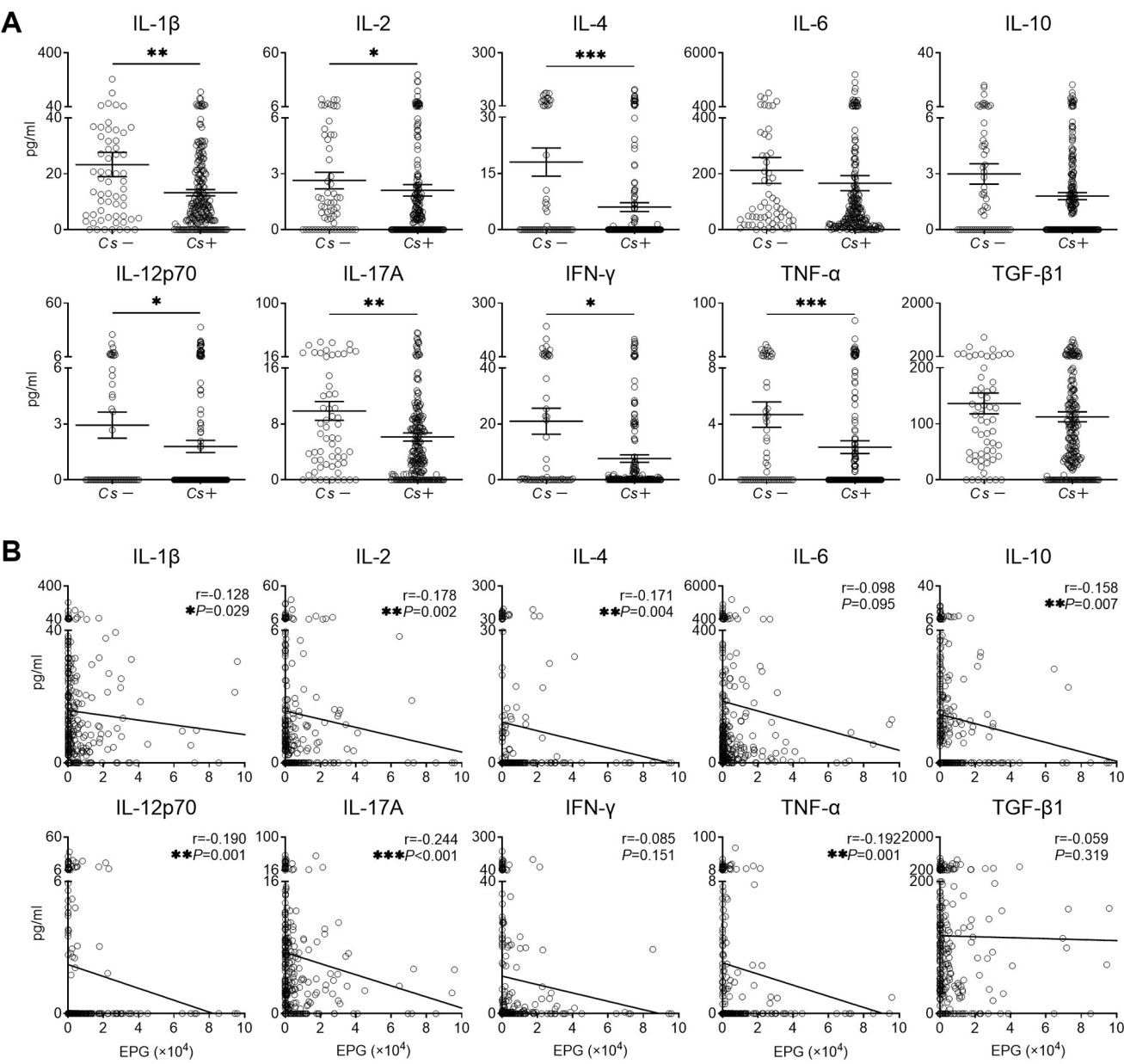

**Fig 3. Serum levels of cytokines in *Cs*- and *Cs*+ individuals. (A)** Serum levels of IL-1β, IL-2, IL-4, IL-6, IL-10, IL-12p70, IL-17A, IFN-γ, TNF-α and TGF-β1 in *Cs*- individuals (n = 60) and *Cs*+ individuals (n = 229). **(B)** Spearman's correlation analysis between EPG and serum levels of IL-1β, IL-2, IL-4, IL-6, IL-10, IL-12p70, IL-17A, IFN-γ, TNF-α and TGF-β1 (n = 289). *Cs*: *C. sinensis* infection, EPG: eggs per gram of feces. The data were shown as the mean ± s. e.m., *$P < 0.05$, **$P < 0.01$, ***$P < 0.001$, Mann-Whitney U test in **(A)**, Spearman's correlation test in **(B)**.

PCA analysis showed very little clustering between *Cs*- male and female population. These findings suggested that host immune status was influenced by sex in *C. sinensis* infection.

## Diminished cytokines in *C. sinensis* infection are not related to hepatobiliary morbidities or soil-transmitted helminth co-infection

*C. sinensis* infection may result in various complications in the liver and biliary systems, which might dampen immune responses. Therefore, we asked whether diminished cytokine profiles

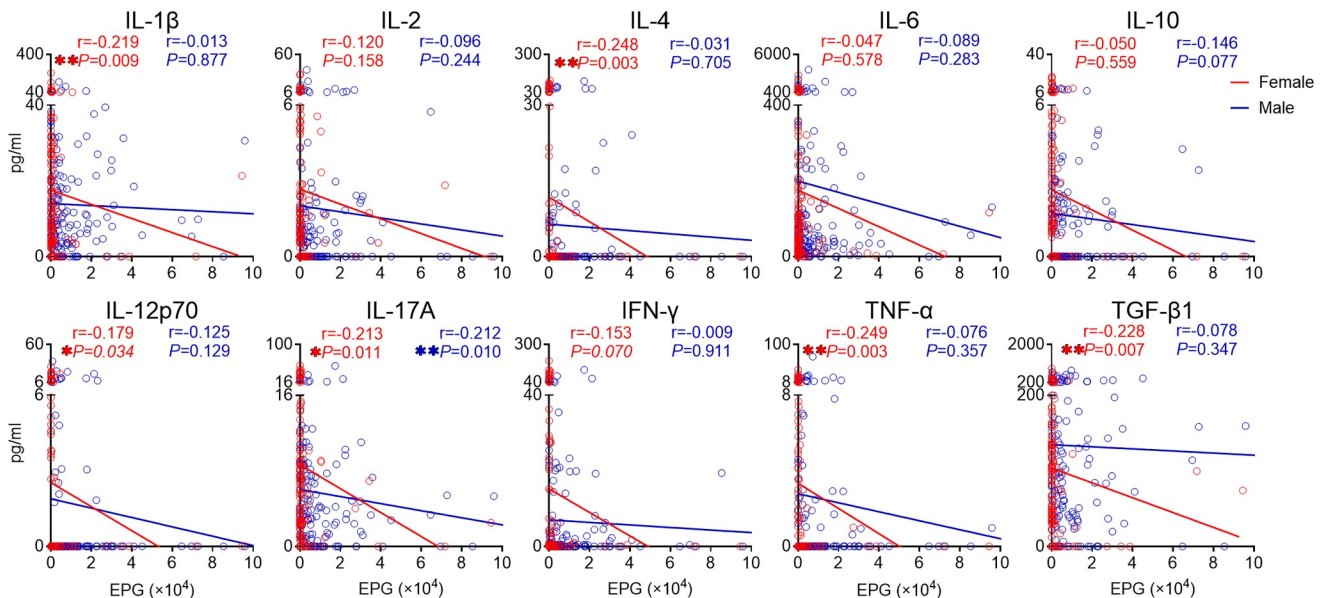

**Fig 4. Spearman's correlation analysis between EPG and serum levels of cytokines in female and male groups.** Spearman's correlation analysis between EPG and serum levels of IL-1β, IL-2, IL-4, IL-6, IL-10, IL-12p70, IL-17A, IFN-γ, TNF-α and TGF-β1 in female (n = 141) and male (n = 148). EPG: eggs per gram of feces. The data were shown as the mean ± s.e.m., $^*P<0.05$, $^{**}P<0.01$, Spearman's correlation test.

in *C. sinensis* infection is associated with hepatobiliary morbidities. Excluding individuals with hepatobiliary diseases detected by ultrasound, the remaining *Cs+* individuals also had significant lower levels of IL-1β, IL-2, IL-4, IL-12p70, IL-17A, IFN-γ and TNF-α in comparison with *Cs-* individuals (Fig 6A). Spearman's correlation analysis indicated that the levels of IL-1β, IL-

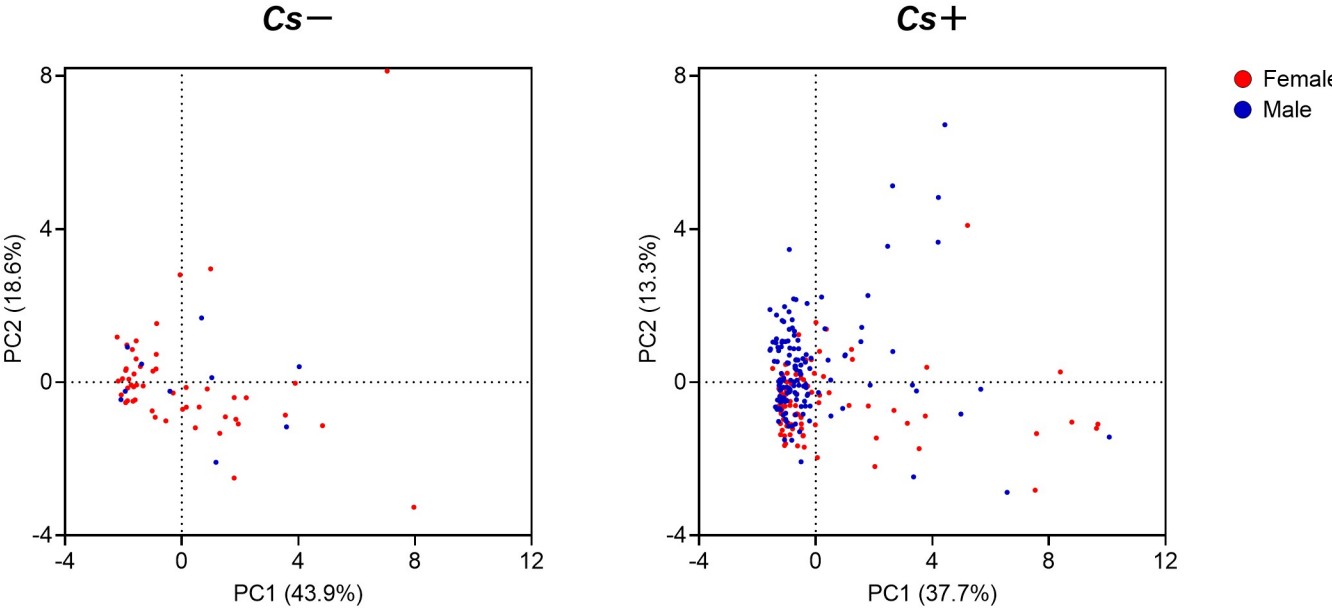

**Fig 5. Principle component analysis (PCA) plots of anti-CsAWA specific antibodies and cytokines from *Cs-* and *Cs+* individuals.** The PCA stands for the two principal components of variation. Left panel, *Cs-* individuals (n = 50 in female and n = 10 in male). Right panel, *Cs+* individuals (n = 91 in female and n = 138 in male). *Cs*: *C. sinensis* infection.

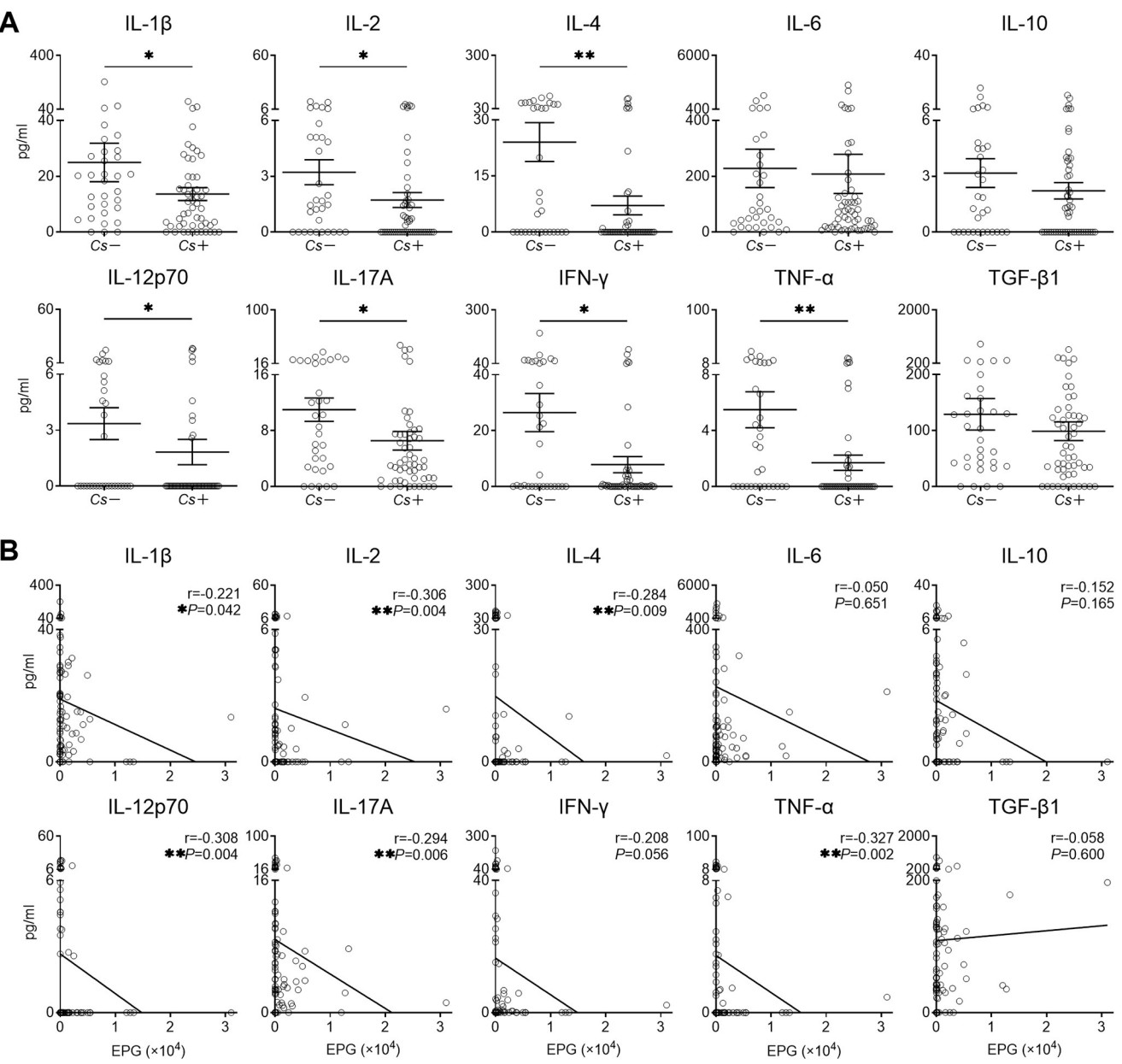

**Fig 6. Serum levels of cytokines in *Cs-* and *Cs+* individuals without hepatobiliary diseases. (A)** Serum levels of IL-1β, IL-2, IL-4, IL-6, IL-10, IL-12p70, IL-17A, IFN-γ, TNF-α and TGF-β1 in *Cs-* individuals (n = 32) and *Cs+* individuals (n = 53) without hepatobiliary diseases. **(B)** Spearman's correlation analysis between EPG and serum levels of IL-1β, IL-2, IL-4, IL-6, IL-10, IL-12p70, IL-17A, IFN-γ, TNF-α, and TGF-β1 (n = 85). *Cs*: *C. sinensis* infection, EPG: eggs per gram of feces. The data were shown as the mean ± s.e.m., *$P < 0.05$, **$P < 0.01$, Mann-Whitney U test in **(A)**, Spearman's correlation test in **(B)**.

2, IL-4, IL-12p70, IL-17A and TNF-α were significantly decreased with increasing EPG in *Cs+* individuals without hepatobiliary morbidities (Fig 6B). In addition, we analyzed the effects of different hepatobiliary diseases on cytokines in *Cs+* individuals respectively. Levels of most cytokines were similar between individuals with and without periductal fibrosis, fatty liver, or bile duct dilatation (S5 Fig). Therefore, diminished serum levels of key cytokines in *C. sinensis* infection were due to *C. sinensis* infection other than the morbidities caused by *C. sinensis* infection.

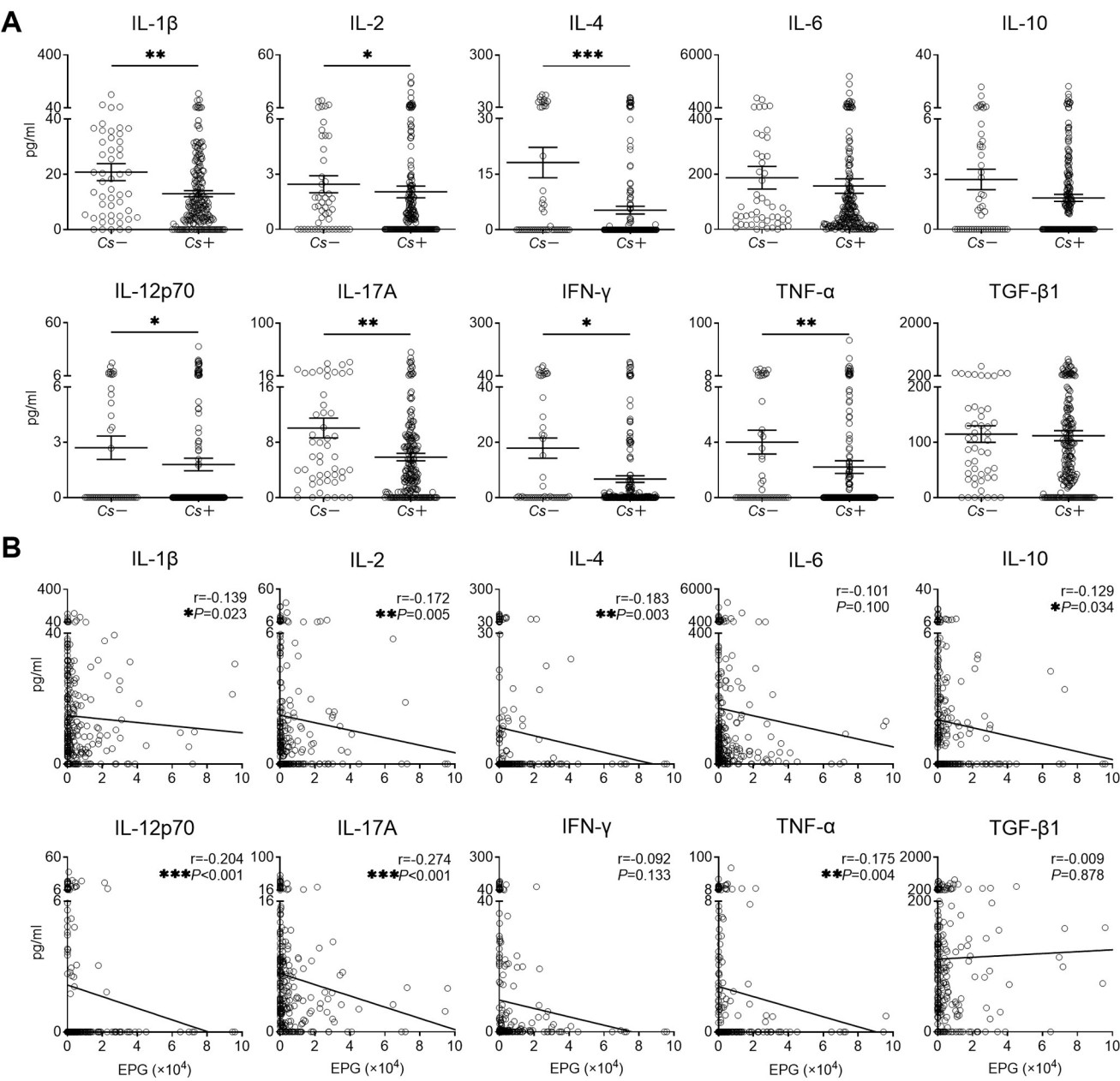

**Fig 7. Serum levels of antibodies and cytokines in *Cs-* and *Cs+* individuals excluding co-infection with soil-transmitted helminths. (A)** Serum levels of IL-1β, IL-2, IL-4, IL-6, IL-10, IL-12p70, IL-17A, IFN-γ, TNF-α and TGF-β1 in *Cs-* individuals (n = 50) and *Cs+* individuals (n = 218) excluding co-infection with soil-transmitted helminths. **(B)** Spearman's correlation analysis between EPG and serum levels of IL-1β, IL-2, IL-4, IL-6, IL-10, IL-12p70, IL-17A, IFN-γ, TNF-α and TGF-β1 (n = 268). *Cs*: *C. sinensis* infection, EPG: eggs per gram of feces. The data were shown as the mean ± s.e.m., * *P*< 0.05, ** *P*< 0.01, *** *P*< 0.001, Mann-Whitney U test in **(A)**, Spearman's correlation test in **(B)**.

Infections with soil-transmitted helminths (hookworms, roundworms, and whipworms) are widespread in tropical and subtropical regions. We collected the information about whether participants have co-infection with soil-transmitted helminths (S1 Table). Excluding co-infection with soil-transmitted helminths, individuals with *C. sinensis* still had significantly higher antigen-specific IgG and IgE levels, and diminished serum levels of IL-1β, IL-2, IL-4, IL-12p70, IL-17A, IFN-γ and TNF-α (Fig 7).

## Discussion

Helminths are extraordinarily successful parasites due to their ability to modulate the host immune response [1,2,5]. Th2 type immune responses are characteristic features of human infection with multicellular parasites [3,19]. In laboratory animals, clonorchiasis, like other helminth infections, is a potent inducer of Th2 responses [8,20,21]. Choi et al. studied antibody and cytokine responses in mice infected with *C. sinensis* and found that susceptibility to *C. sinensis* infection was associated with Th2 cytokine production, especially IL-4 [22]. Wang and colleagues carried out an experimental model in rats and found that immune response presented a tendency to Th2 type by expressing transient high levels of IgG, IgE and IL-4 [23]. Our present study described the immune responses of clonorchiasis in humans. As expected, anti-CsAWA IgG and IgE antibody levels were elevated in individuals infected with *C. sinensis*. Serum IgG and IgE levels were correlated with egg output in the stool, suggesting anti-*C. sinensis* specific immune responses were induced in infected people. However, systemic cytokines represented a general decline in human clonorchiasis. Not only Th1 cytokines (IL-1β, IL-2, IL-12p70, IFN-γ and TNF-α), but also Th2 cytokine (IL-4) and Th17 cytokine (IL-17A) reduced markedly in *C. sinensis* infected people. Negative correlation between cytokine levels and egg counts further supported the characteristics of general immune suppression in *C. sinensis* infection. In this study, *C. sinensis* infection didn't show increased Th2 immune response. It might due to the duration of infection. Acuteness and chronicity of infection drive distinct immune profiles. According to literatures, in *C.sinensis* infected mice, IL-4 production by splenocytes increased (> threefold) until 2–4 weeks post-infection, but declined thereafter [22]. In rat models, compared with control, IFN-γ and IL-4 levels were elevated post infection, and both decreased to lower levels at week 16 after primary infection [24]. In human beings, *C. sinensis* usually causes long-term infection. Untreated, infection may persist for up to 25–30 years [7]. With long-standing chronic infection, immune suppression might be the dominant phenotype in human clonorchiasis. It is interesting to establish a long-standing, persistent infection model (e.g., more than 24 weeks) to track the dynamic changes of cytokines and investigate the intrinsic mechanism. Helminth infections and their components have been shown to have the potential to modulate and attenuate immune responses [25]. *C. sinensis* infection is carcinogenic to human [26–28]. The pathogenic mechanisms include mechanical injury of biliary epithelia by the flukes, immunopathological changes caused by infection-related inflammation, and direct effects of the excretory-secretory products (ESPs) [8,29–31]. Here we propose, besides above 3 mechanisms, *C. sinensis* infection caused immune suppression might also facilitate the transformation and proliferation of the tumor cells.

Numerous investigations have revealed a bias toward males in the susceptibility to and severity of a variety of infectious diseases, especially parasitic diseases [32,33]. In clonorchiasis, the infection rate and intensity in males is usually higher than that in females [7,34]. The provincial level survey in Guangxi demonstrated a prevalence of 14.0% in male and 7.2% in female in 2019. Consistently, the prevalence in the male was around 2 times than that in the female in local survey in Republic of Korea and Vietnam [35,36]. It was believed that the difference between sexes in *C. sinensis* infection is mainly related to dietary customs namely raw-fishing eating behavior [7,37,38]. We focused on the immune alternations in people with *C. sinensis* infection, and demonstrated that *C. sinensis* infection induced immune suppression was influenced by sex. It would be interesting to explore whether this sex-based immune suppression contributes to the concomitant immunity in *C. sinensis* infection as well as its outcomes.

It has been known that sex broadly influences the host immune response [18]. Both genetic and hormonal factors may result in the sex difference of cytokine milieus [39,40]. For example, genes on the X chromosome code for numerous proteins involved in immune processes,

including pattern recognition receptors (PRRs, e.g., *TLR7* and *TLR8*), transcriptional factors (e.g., *FOXP3*) and main members in nuclear factor-κB pathway (e.g., *IRAK-1*and *NEMO*), which are important in immune cell activation and cytokine production [41,42]. Sex hormones can influence the function of host immune cells by binding to specific receptors that are expressed in most immune cells, such as lymphocytes, macrophages and dendritic cells [39]. Moreover, hormone response elements are present in the promoters of several immune genes, thus sex hormones may directly alter gene expression and immune response [40]. It was reported that the expression of PRRs (e.g., TLR4 and TLR9) could be regulated by sex hormone [43]. Innate immune cells from males express higher levels of TLR4 and produce more pro-inflammatory cytokine TNFα and chemokine CXCL10 than female cells both constitutively and following activation [39,44]. Activation of TLR9 in PBMCs from human males results in more IL-10 production compared with cells from females, which is positively correlated with androgen concentration in males [39]. In this study, we found that female individuals developed less specific IgE and had more reduction of systemic cytokines compared to males with similar infection intensity. We focused on the immune profiling and reported a new characteristic of host immune response in clonorchiasis. Due to the problem of insufficient specimen, we have not analyzed sex hormone levels. To further explore the underlying mechanisms of immune suppression in clonorchiasis, the roles of sex hormones are worth to be investigated.

In conclusion, our study demonstrated *C. sinensis* infection is strongly related to the immune suppression in human being and it is influenced by sex. It provides a better understanding of how worms affect immune responses and cause a long-term immune alternation in humans with *C. sinensis* infection. This finding may benefit to the prevention of clonorchiasis and subsequent morbidity. Moreover, the influence of sexual dimorphism is worth to be further explored in clonorchiasis.

## Supporting information

**S1 Table. Characteristics of individuals with or without *C. sinensis* infection.** (XLSX)

**S1 Fig. Serum levels of antibodies in *Cs*- and *Cs*+ individuals with different ages.** Spearman's correlation analysis between EPG and serum levels of anti-CsAWA IgG and IgE (n = 37 in 10–29 years, n = 73 in 30–44 years, n = 97 in 45–59 years, and n = 82 in 60–86 years). CsAWA: *C. sinensis* adult worm antigen, EPG: eggs per gram of feces. The data were shown as the mean ± s.e.m., $^{**}P < 0.01$, $^{***}P < 0.001$, Spearman's correlation test. (TIF)

**S2 Fig. Serum levels of cytokines in *Cs*- and *Cs*+ individuals with different ages.** Spearman's correlation analysis between EPG and serum levels of IL-1β, IL-2, IL-4, IL-6, IL-10, IL-12p70, IL-17A, IFN-γ, TNF-α and TGF-β1 in different age groups (n = 37 in 10–29 years, n = 73 in 30–44 years, n = 97 in 45–59 years, and n = 82 in 60–86 years). CsAWA: *C. sinensis* adult worm antigen, EPG: eggs per gram of feces. The data were shown as the mean ± s.e.m., $^{*}P < 0.05$, $^{**}P < 0.01$, Spearman's correlation test. (TIF)

**S3 Fig. Serum levels of cytokines in females and males without *C. sinensis* infection.** Serum levels of IL-1β, IL-2, IL-4, IL-6, IL-10, IL-12p70, IL-17A, IFN-γ, TNF-α and TGF-β1 in *Cs*-individuals (n = 50 in female and n = 10 in male). *Cs*: *C. sinensis* infection, F: female, M: male. The data were shown as the mean ± s.e.m., Mann-Whitney U test. (TIF)

**S4 Fig. Spearman's correlation analysis between EPG and serum levels of cytokines in female and male groups with different ages.** Spearman's correlation analysis between EPG and serum levels of IL-1β, IL-2, IL-4, IL-6, IL-10, IL-12p70, IL-17A, IFN-γ, TNF-α and TGF-β1 in 30–44 y (n = 38 in female and n = 35 in male), 45–59 y (n = 46 in female and n = 51 in male) and 60–86 y (n = 44 in female and n = 38 in male). EPG: eggs per gram of feces. The data were shown as the mean ± s.e.m., *P< 0.05, **P< 0.01, Spearman's correlation test. (TIF)

**S5 Fig. Serum levels of cytokines in *Cs*+ individuals with or without hepatobiliary morbidities. (A)** Serum levels of IL-1β, IL-2, IL-4, IL-6, IL-10, IL-12p70, IL-17A, IFN-γ, TNF-α and TGF-β1 in *Cs*+ individuals with (n = 159) or without periductal fibrosis (n = 70). **(B)** Serum levels of cytokines in *Cs*+ individuals with (n = 39) or without fatty liver (n = 190). **(C)** Serum levels of cytokines in *Cs*+ individuals with (n = 55) or without bile duct dilatation (n = 174). PF: periductal fibrosis, FL: fatty liver, BDD: bile duct dilatation. The data were shown as the mean ± s.e.m., *p < 0.05, Mann-Whitney U test. (TIF)

## Acknowledgments

Thanks Dr. Deng-Yu Liu from Guangxi Medical University for providing *C. sinensis* adult worm.

## Author Contributions

**Conceptualization:** Men-Bao Qian, Zhao-Jun Wang.

**Data curation:** Shuo Kan, Qi Li, Yan-Hua Yao.

**Formal analysis:** Qi Li.

**Funding acquisition:** Zhao-Jun Wang.

**Investigation:** Shuo Kan, Qi Li, Xin-Yue Du.

**Methodology:** Shuo Kan, Yan-Hua Yao, Chen-Yun Wu.

**Project administration:** Zhao-Jun Wang.

**Resources:** Hong-Mei Li, Xiao-Kui Guo, Men-Bao Qian, Zhao-Jun Wang.

**Supervision:** Guang-Jie Chen, Xiao-Kui Guo, Zhao-Jun Wang.

**Writing – original draft:** Shuo Kan, Qi Li, Men-Bao Qian.

**Writing – review & editing:** Qi Li, Xin-Yue Du, Guang-Jie Chen, Men-Bao Qian, Zhao-Jun Wang.

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
