## [Decision Letter · Decision Letter 0]

12 Jun 2022

Dear Dr. Wang,

Thank you very much for submitting your manuscript "Clonorchis sinensis infection modulates key cytokines for essential immune response impacted by sex" for consideration at PLOS Neglected Tropical Diseases. As with all papers reviewed by the journal, your manuscript was reviewed by members of the editorial board and by several independent reviewers. In light of the reviews (below this email), we would like to invite the resubmission of a revised version that takes into account the reviewers' comments. 

We cannot make any decision about publication until we have seen the revised manuscript and your response to the reviewers' comments. Your revised manuscript is also likely to be sent to reviewers for further evaluation.

Sincerely,

jong-Yil Chai

Associate Editor

Simone Haeberlein

Deputy Editor

Reviewer's Responses to Questions

**Key Review Criteria Required for Acceptance?**

**Methods**

-Are the objectives of the study clearly articulated with a clear testable hypothesis stated?

-Is the study design appropriate to address the stated objectives?

-Is the population clearly described and appropriate for the hypothesis being tested?

-Is the sample size sufficient to ensure adequate power to address the hypothesis being tested?

-Were correct statistical analysis used to support conclusions?

-Are there concerns about ethical or regulatory requirements being met?

Reviewer #1: (No Response)

Reviewer #2: The status or level of immune response may be related to factors such as gender and duration of infection, as the author mentioned in the manuscript. In fact, the age of the patient was the factor most likely associated with the duration of the infection. Therefore, it is suggested that the authors stratified the data collected based on gender and age. Otherwise, some valuable information or features may be annihilated. If the stratified analysis is not possible due to the number of cases, the author should also discuss this issue appropriately.

Reviewer #3: Are the objectives of the study clearly articulated with a clear testable hypothesis stated? Yes ，they are.

-Is the study design appropriate to address the stated objectives? Yes, it is. 

-Is the population clearly described and appropriate for the hypothesis being tested? Yes, it is. 

-Is the sample size sufficient to ensure adequate power to address the hypothesis being tested? Yes, it is.

-Were correct statistical analysis used to support conclusions? Yes, it is ok. 

-Are there concerns about ethical or regulatory requirements being met? Yes, they are.

**Results**

-Does the analysis presented match the analysis plan?

-Are the results clearly and completely presented?

-Are the figures (Tables, Images) of sufficient quality for clarity?

Reviewer #1: (No Response)

Reviewer #2: (No Response)

Reviewer #3: -Does the analysis presented match the analysis plan? Yes it is ok. 

-Are the results clearly and completely presented? Yes, they are. 

-Are the figures (Tables, Images) of sufficient quality for clarity? Most of the figures are OK.

**Conclusions**

-Are the conclusions supported by the data presented?

-Are the limitations of analysis clearly described?

-Do the authors discuss how these data can be helpful to advance our understanding of the topic under study?

-Is public health relevance addressed?

Reviewer #1: (No Response)

Reviewer #2: This study was mainly based on serological data and did not study the mechanism of host immunosuppression caused by C. sinensis infection. Therefore, the expression "This finding expends our understanding of how worms affect immune response" in the abstract and the conclusion of the manuscript is inadequate.

Reviewer #3: -Are the conclusions supported by the data presented? Yes, they are

-Are the limitations of analysis clearly described? Yes, it have been described. 

-Do the authors discuss how these data can be helpful to advance our understanding of the topic under study? Yes, they did. 

-Is public health relevance addressed? Yes, it is.

**Editorial and Data Presentation Modifications?**

Reviewer #1: (No Response)

Reviewer #2: (No Response)

Reviewer #3: Major revision will be better.

**Summary and General Comments**

Reviewer #1: In this study, the authors found the characteristics of host immune response of human infected with C. sinensis. After C. sinensis infection, antigen-specific IgG and IgE levels were higher than control people. In addition, the antigen-specific IgE level had sex difference. C. sinensis infection caused diminished Th1 cytokines, Th2 cytokine, and the changes of cytokine milieus also had sex difference, indicating that the immune response was suppressed and had sexual dimorphism. However, there was no significant correlation between the dampened cytokine profiles and the hepatobiliary morbidities. This study provides a better understanding of how worms affect immune responses and cause a long-term immune alternation in humans with C. sinensis infection. However, there are some problems in the article. The major comments are as follows:

1.The specific IgE level and cytokine milieus had sex differences in C. sinensis infection. PCA analysis indicated that there was a relation between cytokine and antibodies clusters in male and female individuals infected with C. sinensis. Could you explain the factors resulted in the sex difference of cytokine milieus? 

2.Perhaps sex hormones related to the sex differences of specific IgE level and cytokine milieus. The detection of sex hormones was suggested.

3.In the discussion, line 279: Wang and colleagues carried out an experimental model in rats and found that immune response presented a tendency to Th2 type by expressing transient high levels of IgG, IgE and IL-4. In this study, Th2 cytokine (IL-4) reduced markedly in C. sinensis infected people. In the discussion, the low level of IL-4 might relate to the immune suppression caused by long time duration of infection. It is interesting to explore the time of IL-4 level alternation. 

4.In the part of “abstract”, please illustrate the numbers of individuals infected with or without Clonorchis sinensis respectively.

5.There are some mistakes in grammar, please revise them, such as line 69: there is a "," after meanwhile and please revise it to "meanwhile,"; line 87:“because the carcinogenesis in fatal cholangiocarcinoma” turn to “because of the carcinogenesis in fatal cholangiocarcinoma”.

Reviewer #2: In this study, the immune response status of clonorchiasis individuals and uninfected persons was observed and analyzed. The main findings are: the serum specific IgE level of female infected patients was lower than that of male infected patients, and Th1, Th2 and Th17 cytokines were lower and more obvious in females. There was no correlation between cytokine levels and hepatobiliary morbidity. This study has certain significance for further understanding the immunological characteristics of clonorchiasis and the mechanism of worm infection regulating host immune response. The English writing is clear and fluent. However, there are also some obvious deficiencies, and it is suggested that the author make some modifications. 

Besides, Line 110: the sentence “The eggs per gram of feces (EPG) is calculated by the average number of eggs in three Kato-Katz smears by a factor of 24.” Is not correct.

Reviewer #3: This manuscript showed us a interesting story that C. sinensis infection is strongly related to the immune suppression in human and declared that sex differences can shape the immune milieus of clonorchiasis.These results are some interesting. However, we have some comments.

1. The author showed that phenotypes were significantly in female than those in males. However, for detection the sex difference, the results is very limited. Sex difference is better considered the hormonal deffernce from the different age.Wether there exsit some difference at baseline levels from the male and female should be elustrated. And I think if the author can add some animals experiment evidence can be better. 

2. For the cases included in the experiment, it need desrirbed whether these patients have any other diseases or co-infection. 

3. Colors of the pictures had better have consistent.
---

## [Decision Letter · Decision Letter 1]

10 Aug 2022

Dear Dr. Wang,

We are pleased to inform you that your manuscript 'Clonorchis sinensis infection modulates key cytokines for essential immune response impacted by sex' has been provisionally accepted for publication in PLOS Neglected Tropical Diseases.

Best regards,

jong-Yil Chai

Academic Editor

Simone Haeberlein

Section Editor

---

## [Editor Report · Acceptance letter]

25 Aug 2022

Dear Dr. Wang,

We are delighted to inform you that your manuscript, "Clonorchis sinensis infection modulates key cytokines for essential immune response impacted by sex," has been formally accepted for publication in PLOS Neglected Tropical Diseases.

Best regards,

Shaden Kamhawi

co-Editor-in-Chief

Paul Brindley

co-Editor-in-Chief
